# Black Devils in Normandy—Identification of an Unknown Soldier Found in the Polish War Cemetery of Urville-Langannerie (France)

**DOI:** 10.3390/genes14030551

**Published:** 2023-02-22

**Authors:** Dagmara Lisman, Milena Bykowska, Joanna Drath, Grażyna Zielińska, Maria Szargut, Jarosław Piątek, Sandra Cytacka, Joanna Dowejko, Julia Zacharczuk, Jan Ambroziak, Andrzej Ossowski

**Affiliations:** 1Forensic Genetics Department, Pomeranian Medical University, 70-204 Szczecin, Poland; 2The Ministry of Culture Heritage and Sport, 00-071 Warszawa, Poland

**Keywords:** mtDNA identification, heteroplasmy, Black Devils, Normandy

## Abstract

A paper dedicated to the identification of a Polish soldier from the 1st Armoured Division under the command of General Stanisław Maczek, who fell in 1944 in Normandy, during World War II. The remains were found at the Urville-Langannerie Polish War Cemetery. A team from the Department of Forensic Genetics at the Pomeranian Medical University in Szczecin, commissioned by the Ministry of Culture Heritage and Sport, exhumed the remains in order to carry out genetic identification tests. A comprehensive anthropological analysis of the heavily degraded remains was carried out, and biological samples were secured for genetic testing. The identification of Jan Dusza is the first case of restoring the identity of an active combatant from the First Armoured Division. In the case analysis, the analysis of mitochondrial DNA in highly degraded biological material proved crucial. Genetic studies decided to reject the original historical hypothesis No. I at their preliminary stage. Regarding hypothesis No. II, a comprehensive genetic analysis of mitochondrial and autosomal DNA was carried out. Comparative material was obtained from the alleged victim’s sister. Thanks to the analysis of kinship in the maternal line based on the mtDNA haplotype, it was possible to establish that the remains belong to Jan Dusza, who served in the Podhale Rifle Battalion, part of the Polish 1st Armoured Division. The research was co-financed by the Polish Ministry of Heritage and National Culture.

## 1. Introduction

At the Polish War Cemetery in Urville-Langannerie lie nearly 600 soldiers of the Polish Army, mainly from the 1st Polish Armoured Division, most of them killed in August 1944 in Normandy during the so-called *Battle of the Falaise* Pocket [1]. Among them are the burials of 28 people whose identities could not originally be determined, often as a result of the severe fragmentation of the bodies, the high degree of decomposition, the charring of the remains and their transfer from field graves to the war cemetery. Such a large number of unknown soldiers and the need to identify them led the Ministry of Culture and National Heritage (MKiDN) to collaborate with the Department of Forensic Genetics (ZGS) of the Pomeranian Medical University (PUM) in Szczecin in a project aimed at restoring the identity of all those buried in unmarked graves at Langannerie. The research undertaken, exhumation and identification of the remains found in grave No. II D 9 precisely captures the essence of this interdisciplinary project.

Soldiers of the Polish Armed Forces in the West fought within the structures of Allied units from the beginning of the Second World War. At the opening of the second (western) frontline in Normandy, the Polish 1st Armoured Division, commanded by Brigadier-General Stanislaw Maczek, was part of the Canadian II Corps of the Canadian 1st Army, in August 1944.

The First Armoured Division, nicknamed the ‘Black Devils’, played a key role in closing the trap at Falaise, which broke through the German defences in Normandy. The assault began on 7 August 1944, and the bloody battle continued for fourteen more days. The signal for the attack by the First Armoured Division came when British and American bombers struck German positions. However, there was a mistake whereby instead of bombing the Germans, the Allied aviation attacked the Polish forces. It was as a result of these actions that the soldier under examination was killed. Within the first week, the entire II Corps had advanced only about 10 kilometres. The turning point in breaking through the German defences occurred on 15 August, when the 10th Horse Rifle Regiment led to the capture of the German crossing of the Dives River. After two days, two key points of the Battle of Falaise, the town of Chambois and the Mont Ormel hill complex near the village of Langannerie, were captured by the Polish unit, which fought bloody battles to hold the position cutting off the Germans from their retreat route in front of the Allied forces. The holding of the position off the hills of Mont Ormel cost the Poles nearly 700 casualties [1].

The analysis of historical data allowed us to hypothesise that the remains found belong to lancer Michał Mosur, killed as a result of the bombing of 14 August 1944. We had comparative material from two nephews of the alleged victim. However, this hypothesis was falsified by analysis of autosomal STR markers and the Y chromosome haplotype at the preliminary stage of genetic research. This caused the list of victims to be re-verified. Attention was drawn to the Podhale Rifle Battalion, where the fate of one of the fallen soldiers remained unknown. A report found in the Central Military Archives in Warsaw allowed Hypothesis II to be put forward. It was assumed that the exhumed remains might belong to Corporal Jan Dusza. This had to be confirmed by genetic research, which in this case also proved irreplaceable. Comparative material was obtained from the sister of the fallen soldier, and then a comprehensive analysis of nuclear and mitochondrial DNA was carried out from the secured biological samples. This allowed Hypothesis II to be verified and the remains of a hitherto unknown man to be identified.

### Purpose of the Work

The main aim of the study was to develop an interdisciplinary procedure model for the identification of Polish soldiers killed abroad on the frontlines. This procedure was to include stages from historical analyses, through anthropological methods, to comparative genetic research. The model thus constructed was to be verified during the identification of an NN male whose remains were deposited at the Polish War Cemetery in Grainville-Langannerie.

The specific aim of the study was to verify in detail the molecular biological methods concerning the mode of treatment of fallen soldiers of the Polish Armed Forces in the West.

## 2. Material and Methods

The exhumation site—the Polish War Cemetery in Grainville-Langannerie (Figure 1)—was selected on the basis of a search and analysis of historical sources held at the Central Military Archives and the Archives of the Commonwealth War Graves Commission, made available by the Polish Ministry of Culture and National Heritage (MKiDN) for the purposes of this research.

A grave cavity discovered during exhumation work revealed human remains (Figure 2), transferred to the Polish War Cemetery in Grainville-Langannerie from the original burial site.

The exhumed remains bore the signs of injuries characteristic of an explosion (multifracture bone fractures, burned torso bones, destroyed skull structure) and were in poor condition, highly degraded and defragmented. This may have been mainly due to their handling and the decomposition that occurred in the wooden coffin.

The state of preservation of the remains, the characteristic injuries and the fact that the remains were buried in a coffin indicated that the DNA contained in the material could also be characterised by a high degree of degradation. Based on our own experience and the results of published work, DNA tests from the skeleton secured the shaft of the right femur, the scaled part of the right temporal bone and the teeth of the left maxilla (from I^1^ L to M^3^ L).

For the purpose of personal identification, comparative material—an oral swab—was also taken for testing from the genetically closest relatives of the missing persons.

## 3. Anthropological Methods

### 3.1. Biological Profile Methodology

The biological age was estimated based on the ossification level of skeletal elements [2], the auricular surface morphology [3], and the teeth eruption [4] and attrition [5]. The biological sex was estimated based on the measurements of the glenoid fossa and the diameter of the acetabulum [6]. The stature was estimated based on the femur measurements with the use of selected formula considering the sex and population (inferred from historical sources and the targeted people of the missing persons list) [7].

In trauma analysis, the anthropological definitions of the terms perimortem, antemortem and postmortem were applied (perimortem trauma is any trauma inflicted to fresh, hydrated bone).

### 3.2. Genetic Methods

#### 3.2.1. DNA Extraction

The evidence secured was cleaned and crushed according to the procedure developed by the team and described in [8]. DNA from the evidence was isolated using the PrepFiler^®^ BTA Forensic DNA Extraction Kit (Applied Biosystems Waltham, Massachusetts, Stany Zjednoczone) according to the protocol [9], using 50 mg of bone powder each time. DNA isolation from reference material was performed using the PrepFiler^®^ Forensic DNA Extraction Kit (Applied Biosystems) according to the manufacturer’s protocol.

#### 3.2.2. DNA Concentration Measurement and Inhibition Evaluation

The Quantifiler Human DNA Quantification Kit (Applied Biosystems) was used to assess the concentration of human DNA and the presence of PCR inhibitors [10], along with the 7500 Real-Time PCR System thermocycler (Applied Biosystems). C_T_ ≥ 31 was used as the criterion for the presence of PCR inhibitors, according to the manufacturer’s instructions and internal method validation.

#### 3.2.3. STR Amplification and Product Detection

Autosomal STR markers were amplified using the GlobalFiler™ PCR Amplification Kit (Thermo Fisher Scientific), and for amplification of Y-STR markers the Yfiler™ Plus PCR Amplification Kit (Thermo Fisher Scientific) was used. Reactions were performed according to the manufacturer’s instructions.

Product detection was performed on a 3500 Genetic Analyzer sequencer using the GeneScan™ 600 LIZ^®^ size standard, according to the manufacturer’s protocol, and the results were then analysed using the GeneMapper ID-X v1.6 software, as in [11].

#### 3.2.4. mtDNA Analysis

Two hypervariable fragments of the mtDNA control region (HV1 and HV2) were sequenced, after which the results were related to the revised Cambridge Reference Sequence (rCRS) using the Sequencher 5.4.6 program (Gene Codes) according to a previously published protocol for the analysis of degraded genetic material (Skeletal evidence of the ethnic cleansing actions in the Free City of Danzig (1939–1942) based on the KL Stutthof victims analysis).

#### 3.2.5. Kinship Testing

The results of the genetic analyses were used to verify Hypotheses I and II by determining the likelihood ratio (LR) of the assumed relatedness using the GenoEvidence 3 software (qualitype GmbH), where the null hypothesis was considered to be the relatedness mapped on Figure 1 and Figure 2 (below), and the alternative hypotheses were that the NN remains belonged to an unrelated male from the population. In addition, the concordance of Y-STR and mtDNA haplotypes between the evidence and reference material was also tested to enhance the discriminatory power of the tests performed for Hypotheses I and II, respectively.

In view of the results of the historical analyses, an initial hypothesis (Hypothesis I) was put forward that the NN male was Michał Mosur. In order to verify the hypothesis, comparative material was taken from the closest relatives of Michał Mosur in the male line—his two nephews (Figure 1).

As Hypothesis I was rejected by genetic analysis (see Section 4 below), a further search was undertaken, followed by Hypothesis II, that the exhumed remains might belong to Jan Dusza; in order to verify the hypothesis, material was secured from his sister (Figure 2).

## 4. Results

### 4.1. Anthropological Results

The skeletal remains belong to a male individual measuring 166–173 centimetres in height who was aged 20–34 years at the time of death (with skeletal indicators like teeth attrition and vertebral rings ossification suggesting biological age under 30 years old, whereas auricular surface morphology showed an older biological age).

Perimortem traumatic injuries resulting in comminuted fractures were visible on cranial bones, as well as both humeri, both forearm bones from both sides, left and right tibias and left femur. Right femur and a piece of cranial vault manifested imbedded shrapnel pieces. The overall trauma pattern, including fire trauma, indicates a blast event responsible for bone fractures and the possible cause of death. The evidence could also point to a close proximity of the person to the epicentre of the explosion.

Anthropological methods in this case proved insufficient to make an identification; however, the analysis of the peri-mortem injuries made it possible to give direction to further research and to draw conclusions leading to hypotheses regarding the potential identity of the male John Doe.

### 4.2. Genetic Testing Results

Two of the three isolates showed the presence of human DNA at a level that allowed STR analyses. No human DNA was detected at a quantifiable level in a sample from the scala of the right temporal bone; no PCR inhibitors were detected in any of the samples (Table 1).

Analysis of the electropherograms derived from STR and Y-STR markers in the evidence samples indicates degradation of the genetic material contained in the sample, as evidenced by the decreasing height of the peaks as the length of the examined fragment increases (due to the protection of personal data of the victim’s relatives (data not shown). Based on the results of capillary electrophoresis, a consensus profile was constructed for both autosomal STR and Y-STR markers, which was judged to be suitable for comparative studies. This result demonstrates the effectiveness of the genetic methods used to obtain data that allow comparative studies to be carried out even when using degraded bone material.

Verification of Hypothesis I

Due to the rather distant consanguinity between Michal Mosur and the men from whom material was secured to verify Hypothesis I, allelic concordance between the subjects was not tested for STR markers, and only the LR value for the assumed consanguinity was determined (Figure 3).

The results of the calculations give a moderately strong indication that the man whose remains were identified is not the brother of the father of Michał Mosur’s nephews. The results of the Y-STR haplotype concordance test further ruled out a close relationship between Michal Mosur’s nephews and the male NN identified (Table 2).

On the basis of the results obtained, Hypothesis I (that the identified man would be Michal Mosur) was definitively rejected.

Verification of Hypothesis II

Allelic concordance between the NN and Jan Dusza’s sister is included in Table 3.

Due to the Mendelian mechanism of inheritance of the studied genetic traits, the lack of concordance between NN male and Jan Dusza’s sister in the three STR systems did not allow the rejection of Hypothesis II; indeed, it further indicated the possibility of consanguinity between the studied individuals. In view of the above, the LR value for the assumed relatedness was calculated (Figure 4).

The LR value for the presumed case gives a very strong indication that the man studied is the brother of Jan Dusza’s sister. In addition, the results of the mtDNA haplotype analyses based on the HV1 and HV2 fragments showed an identical haplotype for the NN of the man and the sister of Jan Dusza (Table 4).

Analysis of polymorphic mitochondrial DNA fragments in both the evidence sample D1 and the reference sample P3 yielded an identical haplotype that did not appear in the database among the 12,795 samples collected from individuals representing the resident population (EMPOP mtDNA database, v4/R13, accessed at empop.online on 12 May 2022). Based on this detailed calculation, the maximum haplotype frequency in the population is 2.88 × 10^−4^.

In the evidence sample, the occurrence of length heteroplasmy in the HVR2 region was additionally noted. The occurrence of heteroplasmy is a fairly common phenomenon. Its level in individual tissues may vary, but it is nevertheless not a decisive feature for rejecting or confirming the hypothesis of consanguinity between two individuals under investigation [12].

Statistical analyses, taking into account the combination of STR markers and the mtDNA haplotype of the evidence D1 and reference P3 samples, made it possible to conclude that the studies obtained give an extremely strong indication that the male NN is the brother of Jan Dusza’s sister.

The results of all the genetic tests carried out on the basis of hypotheses derived from historical analyses and anthropological studies made it possible to identify the remains as belonging to **Corporal Jan Dusza** (Figure 5).

## 5. Discussion

Molecular biology methods used in the identification of missing persons should be routinely applied to the analysis of skeletal remains. The last two decades show that analysis of STR loci is the primary method for identifying human remains [13,14,15,16]. Research to date indicates that, in the case of World War II victims, DNA analysis has become highly effective and, in most cases, the only appropriate method of identification [14,17,18,19,20]. Identification studies of Bosnian and Herzegovinian [21] and Croatian [22,23] war victims, analysis of World War II remains in Slovenia [24,25] and the numerous identification studies carried out by our team [14,15,16,17,26] indicate that components such as the optimisation of DNA isolation protocols and the selection of appropriate testing methods depending on the comparative material collected from the relatives of the victims are extremely important in the restoration of identity. The importance of the very process of DNA isolation from highly degraded material was demonstrated by researchers analyzing ancient DNA [27]. They showed that the methods of dealing with fossil DNA are equally effective and should be used in the identification of historical figures and victims of the armed conflicts of the 20th century.

It has been proven many times that mitochondrial DNA performs well with highly degraded material. This is determined by its characteristics, which also prove to be extremely important for forensic analyses (lack of recombination, matrilineal inheritance, heteroplasmy, high copy number), which is important in the case of time-treated remains. However, it should be borne in mind that the confirmation or exclusion of identical mtDNA sequences refers to a group of individuals related in the same maternal line [28]. The choice of the appropriate identification method is determined by the degree of relatedness. The same is true for the phenomenon of heteroplasmy observed at the same nucleotide positions of the reference and evidence samples. This phenomenon does not allow affinity to be ruled out. Several researchers have suggested that mtDNA samples with a single base difference should still be analysed, taking into account their mutation rate [29,30,31,32]. In our work, the phenomenon of length heteroplasmy was detected in the HV2 region of the evidence sample. The heteroplasmy itself can manifest itself in various ways [33]. The phenomenon of heteroplasmy of the mtDNA control region became crucial in the identification studies of the remains of Tsar Nicholas II of Russia and his brother Grand Duke Georgi Romanov of Russia [34,35]. Other researchers have detected heteroplasmy of two different mtDNA positions of plucked hairs from an anonymous donor. These analyses revealed mtDNA heteroplasmy in blood samples and buccal mucosal swabs from a family (mother and two children) [36]. This is therefore a relatively common phenomenon and, as other studies have shown, may not be decisive in rejecting the hypothesis of consanguinity between the two individuals studied [28].

Another example is the identification of victims of the Spanish War of 1936–39. The remains were exhumed from a mass grave, and mtDNA profiles were matched from the youngest biologically exhumed skeleton and the sister of the youngest person allegedly buried in the mass grave [37,38]. The identification of the remains of an Italian World War I soldier, killed in battle on the Italian frontline in 1915, similarly to our team’s identification study, led to interesting conclusions. The researchers sequenced the HV1/HV2 regions of the mtDNA to analyse its individual haplotypes. Offspring from the mother’s side were used as reference material. The study ruled out that the remains found belonged to the alleged war hero [39]. Our team faced a similar problem. Before the exhumation, it was typified that the remains belonged to another person. Through analysis of the lists of those killed at the time, and given that the only known fallen soldier from the above-mentioned unit who had not as yet received a named burial was lancer Michal Mosur, also killed as a result of the bombardment on 14 August 1944, Hypothesis I was accepted—that his remains were buried in the grave we exhumed [40]. It was possible to obtain comparative material from the alleged nephews of Michal Mosur; they were the victim’s closest living relatives. In this case, the first stage of genetic testing (Y-STR markers) ruled out kinship. These results raised the question of to which of the fallen soldiers the exhumed remains belonged. Again, historical sources were verified. This caused our team to search for the fallen among other units. From the documents analysed, it appeared that eight soldiers from the Podhale Rifle Battalion had fallen on that day, while the fate of one remains unknown. One report suggested the identity of Corporal Jan Dusza, killed on 14 August 1944. The description also found information about the injuries he sustained, which coincided with anthropological studies of the exhumed skeleton. The circumstances of death, as well as the injuries sustained, which pointed to an explosion as the cause of death, also appeared to be the same for both victims. Anthropological methods of identification proved impossible due to the degree of fragmentation of the skeleton. Genetic analyses were necessary. With comparative material from Jan Dusza’s alleged sister, we were able to lead to the identification of Corporal Jan Dusza. A mitochondrial DNA analysis proved necessary, which confirmed the identity of the victim. This is further evidence that genetics remains the gold standard for identification research. Identification should never be based on just one method. Our work shows how easy it is to get it wrong by relying only on the analysis of historical sources and anthropological methods. In both cases, the circumstances of the victims’ deaths, the age at which they died and the place of burial were found to be the same. The gold standard, however, turned out to be genetics.

Our team has extensive experience in identification research. The project of the Polish Genetic Database of Victims of Totalitarianism, which we have been carrying out for years, has made us realise how many people lie in nameless graves and how necessary such research is [16,17,18,24,41]. While conducting exhumation work of a mass grave in the Powązki cemetery in Warsaw, we identified the victims buried in one of the many mass graves by analysing mainly autosomal Y-STR and sequencing HV1/HV2 fragments of the control region of mtDNA [18]. Moreover, in the study presented in this thesis, we benefited from our previous experience. Even though we had excellent historical documentation in this case, we still found that historical, anthropological and medico-legal analyses alone were not sufficient to complete the identification research process.

Today’s state of research shows that haploid markers such as mtDNA have become a useful tool in identification and forensic studies. It is its polymorphic nature and inheritance in the maternal line, combined with the analysis of its sequence, that has allowed successful identification studies, including in our work. Comparative analysis of the mitochondrial DNA sequence of the evidence sample and the reference sample is becoming an extremely valuable tool in human identification, particularly in older cases where we have limited access to close relatives and are dealing with highly degraded material. Success in this process consists of many elements. It involves interdisciplinary cooperation: verification of historical data, reaching out to victims’ relatives, witness statements, medical-forensic examination and anthropological methods. The archaeological and anthropological research carried out is usually based on the analysis of archival data concerning information about the burial place, the circumstances of death and information about the missing person. Most of the material from that period has been destroyed. This is why so many remains cannot be found, nor their identity restored. Another problem is the acquisition of comparative material. This is made more difficult by the time that has elapsed since the death. These difficulties are also encountered by other researchers [22]. The poor state of preservation of the skeleton poses problems in obtaining a good quality DNA matrix, as has been shown in other studies [42,43].

It is worth paying attention to how the exhumation material is preserved. Our experience shows that it is extremely important to avoid contamination and further degradation of the DNA matrix. We know very well that the material for genetic research should be secured immediately after the exhumation of the skeleton, so as not to expose it to further degradation during the process of transport and storage of the entire remains. We set up a medical base at the site of our work, where we carry out preliminary anthropological and medico-forensic examinations. Material for genetic research is stored in sterile phalcones and immediately frozen at -20 ºC until isolation. This procedure is also used by other research teams [44,45]. This concerns the effects of environmental, biological and physical factors on the DNA until transport to the laboratory. Experience has shown us that the freezing process is the simplest protective procedure because the low temperature protects against microbial growth [17]. Similar conclusions have been reached by other teams [46].

The creation of the Polish Genetic Database of Victims of Totalitarianism project involves years of work with highly degraded bone material and the development of appropriate methods of procedure. The team includes historians, forensic anthropologists, archaeologists, forensic medics and forensic geneticists. It is precisely the interdisciplinarity of the research that other teams also prioritise [37].

DNA analyses in identification cases from decades ago are becoming the gold standard, as they are in contemporary cases. However, interdisciplinary cooperation between historians, anthropologists, forensic medics and geneticists is needed for this to happen. The work shows how important the multidisciplinarity of these teams is. Similar conclusions were drawn by researchers analyzing mass graves in Bosnia and Herzegovina [21]. Work carried out by ICMP (The International Commission on Missing Persons) and similar organisations, Polish Genetic Database of Totalitarian Victims (PBGOT) [16], International Committee of the Red Cross (ICRC) and European Union Rule of Law Mission in Kosovo (EULEX), have shown that DNA analyses can be considered as the primary method of identification. Data obtained by DNA typing are considered highly reliable.

## Data Availability

Not applicable.

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
