# Peer review of "Black Devils in Normandy—Identification of an Unknown Soldier Found in the Polish War Cemetery of Urville-Langannerie (France)"

_genes, 2023, doi:10.3390/genes14030551_

Round 1
Reviewer 1 Report
The work proposed is well defined and well presented.
Introduction explains properly the historical data concerning the soldier, the discovery and the retrieval.
Antropological, genetic and statistical methods are deeply and clearly explained.
Results appear well reported and discussed.
Conclusions seem to be appropriated and correct.
Author Response
Thank you for your time and valuable comments.
Reviewer 2 Report
I have just one recommendation regarding proper citing. Please include the papers resulting from the work of ICMP in Bosnia and Herzegovina plus discuss the recommendations given by HOFREITER, Michael, et al. Progress in forensic bone DNA analysis: Lessons learned from ancient DNA. Forensic Science International: Genetics, 2021, 54: 102538.
Author Response
Thank you for your time and valuable comments.
I followed the guidelines. I made changes in the discussion in paragraphs 1 and 8. I included the proposed publication and discussed Hofreiter's guidelines, which were consistent with the main purpose of the publication.
Reviewer 3 Report
After examining the scientific study, the following considerations may be made. The scientific study is well-structured in all its parts. In particular, the premises with which the authors introduced the analysis are clear. Equally, clear are the objectives that led the authors to carry out this study and the section on materials and methods. Particular appreciation can also be expressed of the material on which the study was carried out. The data was collected methodically and without bias. The results were consistent and significant and allowed a discussion section full of food for thought. The authors then developed a discussion of the results achieved.
English is well-structured in syntax and grammar.
Author Response

(The authors gave the same response as above.)

Reviewer 4 Report
1. This article is sound and clearly written.
2. It is better to put tables 2 and 3, 4 and 5 in the same order.
3. To ensure that photo 3 is published with the family's consent.
Author Response
- Thank You.
- Tables 2 and 4 are GenoEvidence 3 screenshots, they are classified as figures because they cannot be edited.
- Of course, we have the consent of the family. The family was informed about the impounded publication and familiarized themselves with its content.